# Vortex knots in tangled quantum eigenfunctions

Alexander J. Taylor[1] & Mark R. Dennis[1]

Tangles of string typically become knotted, from macroscopic twine down to long-chain macromolecules such as DNA. Here, we demonstrate that knotting also occurs in quantum wavefunctions, where the tangled filaments are vortices (nodal lines/phase singularities). The probability that a vortex loop is knotted is found to increase with its length, and a wide gamut of knots from standard tabulations occur. The results follow from computer simulations of random superpositions of degenerate eigenstates of three simple quantum systems: a cube with periodic boundaries, the isotropic three-dimensional harmonic oscillator and the 3-sphere. In the latter two cases, vortex knots occur frequently, even in random eigenfunctions at relatively low energy, and are constrained by the spatial symmetries of the modes. The results suggest that knotted vortex structures are generic in complex three-dimensional wave systems, establishing a topological commonality between wave chaos, polymers and turbulent Bose–Einstein condensates.

[1] H H Wills Physics Laboratory, University of Bristol, Tyndall Avenue, Bristol BS8 1TL, UK. Correspondence and requests for materials should be addressed to A.J.T. (email: alexander.taylor@bristol.ac.uk) or to M.R.D. (email: mark.dennis@bristol.ac.uk).

Complexity in physical systems is often revealed in the subtle structures within spatial disorder. In particular, the complex modes of typical three-dimensional (3D) domains are not usually regular, and at high energies behave according to the principles of quantum ergodicity[1,2]. The differences between chaotic and regular wave dynamics can be seen clearly in the Chladni patterns of a vibrating two-dimensional plate[2], in which the zeros (nodes) of the vibration accumulate small sand particles to appear as random curved lines when the plate is irregular. The spatial distribution of these nodal lines is statistically isotropic at high energies[2], and this irregularity is typical for chaotic systems, whose ergodic dynamics are determined only by the energy and there are no other constants of motion.

Understanding the spatial structure of these wavefunctions can be challenging. Following the hydrodynamic interpretation of single-particle quantum mechanics[3], the zeros of 3D complex-valued scalar fields are in general lines; vortex filaments around which the phase, local velocity and probability current (in a quantum wavefunction) circulate[4–6]. This is analogous to the vortices of a classical fluid, although the phase change around the vortex line is quantized in units of $2\pi$, and is singular at the vortex core where the amplitude is zero. A similar vortex topology occurs in condensates of many quantum particles[7]. The pattern of their vortex lines provides a structural skeleton to wavefunctions[4,6,8]. In modes above the lowest energies, the vortex pattern is far from regular and they are densely intertwined. A natural 3D measure of the most extreme spatial irregularity is when the vortex filaments are knotted, and here we study the occurrence of knotted nodal vortex lines in three model systems of wave chaos, as a natural extension of the Chladni problem to three dimensions.

Despite numerous investigations of the physics of diverse random filamentary tangles, including quantum turbulence[9], loop soups[10], cosmic strings in the early universe[11] and optical vortices in 3D laser speckle patterns[12], the presence of knotted structures in generic random fields has not been previously emphasized or systematically studied. Vortices and defects which are knotted have been successfully embedded in a controlled way in various 3D fields, such as vortex knots in water[13], knotted defects in liquid crystals[14,15] and knotted optical vortices in laser beams[16], and theoretically as vortex lines in complex scalar fields, including superfluid flows[17], and superpositions of energy eigenstates of the quantum hydrogen atom[18] and other wave fields[19], but rigorous mathematical techniques to resolve the statistical topology of random fields are limited[20]. With modern high-performance computers, the structure can be explored using large-scale simulations.

In the following, we investigate the knottedness of the nodal vortex structures in typical chaotic eigenfunctions of comparable size (that is, energy and total nodal line length) for three model systems. The chaotic eigenfunctions are represented as superpositions of degenerate energy eigenfunctions weighted by complex, Gaussian random amplitudes; such superpositions are established as good models of wave chaos in the semiclassical limit of high energy[1,2,21], since the wave pattern is determined only by their energy (unlike a plane wave which has a well-defined momentum). Our three systems are the cubic cell with periodic boundary conditions, whose degenerate eigenfunctions are plane waves; the abstract 3D sphere (3-sphere), whose degenerate eigenfunctions are hyperspherical harmonics and which has finite volume but no boundary; and the isotropic, 3D harmonic oscillator (3DHO), whose nodal structure is largely contained within the classically allowed region, where the energy exceeds the potential such that it could describe an isotropic, harmonically trapped Bose–Einstein condensate[22]. Further

information and illustrations of these eigenfunction systems appear in Supplementary Note 1 and Supplementary Figs 1–4.

We find that vortex knots occur with high probability at sufficiently high energy in all of these random wave superpositions. The results suggest that even in low-energy cavity modes, possibly accessible to experiments, knotted vortices will occur with some reasonable probability. The statistical details of random vortex knotting (with respect to the types of knot that occur, the length of knotted vortex curves and the eigenfunction energy) depends strongly on the wave system, and we perform an analysis and comparison of these properties.

## Results

**Knots in high-energy random eigenfunctions.** Figure 1 shows the nodal/vortex structure of a typical chaotic eigenfunction in each model system: in Fig. 1a, a cube with periodic boundaries; in Fig. 1b, the 3-sphere; and in Fig. 1c, the 3DHO. In all three cases the random modes are labelled by energy $E_N$ with principal quantum number $N$ (further details of how the model systems are generated and how the vortices are located is given in Supplementary Notes 1 and 2). In each of the Figs 1d–g, a single vortex line from these eigenfunctions is shown alongside a simpler projection of the same knot. Each vortex line resembles a random walk[12,23].

Our analysis of these knots uses the standard conventions of mathematical knot theory[24], in which knots in closed curves can be factorized into prime knots, which are tabulated according to their minimum number of crossings (examples are shown in Supplementary Figure 5). Both prime knots (for example, as shown in Fig. 1d,f–g), and composites of prime knots (for example, as shown in Fig. 1e), are identified in the curve data by a combination of several schemes. After simplifying the longest curves by a geometric relaxation method[23], several topological invariants are computed for each curve, which distinguish knots of different types. These are, the absolute value of the Alexander polynomial[25] $|\Delta(t)|$, evaluated for $t$ at certain roots of unity; the hyperbolic volume[26]; and the second- and third-order Vassiliev invariants[27,28]. Although other knot invariants can have more discriminatory power than these separately[24,29], they tend to be significantly more demanding in computer time; in practice the combination of this particular set of invariants discriminates almost all tabulated knots[24,29] (up to at least 14 crossings; the methods are described further in Supplementary Note 3). Each of these invariants encodes different topological and geometric information about each knot[24]. When comparing the knottedness of different vortex loops, we follow previous studies[25,27] in using the (positive integer-valued) knot determinant $|\Delta(-1)|$ as the primary quantitative measure of their knotting complexity.

We have identified $\sim 50{,}000$ knots from around $10^9$ curves (of different lengths), computed from about $10^6$ random eigenfunctions (at different energies). Some of these knotted curves are relatively short, such as that in Fig. 1d, although the majority of vortex knots are much longer, such as that in Fig. 1e, whose full spatial extent spreads over several periodic cells. At large lengthscales, each curve approximates a Brownian random walk[12,23]; for closed curves, the radius of gyration scales with the square root of the total length. Most of the vortex knots occur in closed loops; in the 3DHO, however, most vortex lines stretch to spatial infinity as almost-straight lines in the classically forbidden region. Most of the knots in the 3DHO are in these open curves, such as that represented in Fig. 1g. In the periodic cube, many of the longer lines wrap around the cell a non-zero number of times (that is, they have nontrivial homology, so in a tiling of the 3D cells they would be infinitely long and periodic[12]). Since knots in such a periodic space have not been adequately classified mathematically[24], we only consider knots in closed loops with trivial homology in this system.

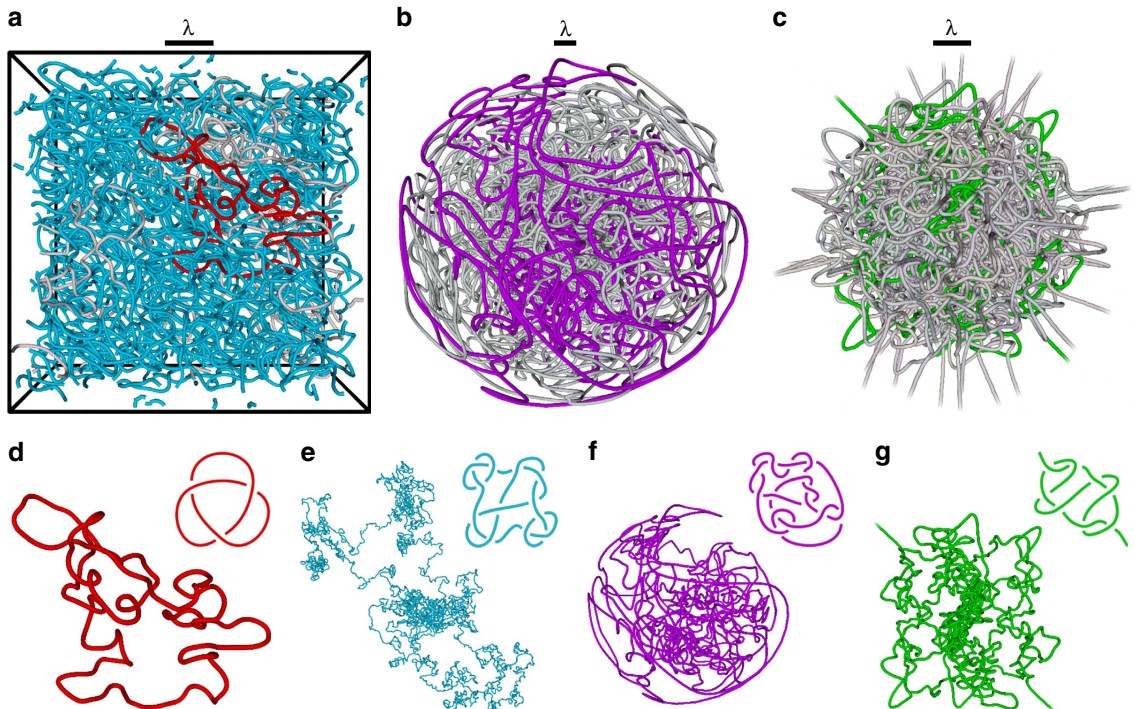

**Figure 1 | Tangled and knotted vortex filaments in random high-energy eigenfunctions of energy $E_N$.** Vortex lines are shown in (**a**) a periodic cubic cell, with principal quantum number $N = 9$ ($E_N \propto 3N^2$); (**b**) the 3-sphere (plotted in a distorted projection in which all points on the bounding sphere are equivalent) with $N = 17$ ($E_N \propto N(N+2)$) and (**c**) the 3DHO with $N = 21$ ($E_N \propto N + 3/2$). The total vortex length is similar in each eigenfunction, a reference wavelength at the origin, proportional to $E_N^{-1/2}$ is given in each of **a**–**c**. Each vortex loop in the eigenfunction is coloured grey except for one or two knotted examples in each system, illustrated further in **d**–**g**; each of these coloured knots is plotted alongside a simpler projection of the same knot. (**d**) The trefoil knot (tabulated as $3_1$ from (**a**) with length $L = 50\ \lambda$ and determinant $|\Delta(-1)| = 3$; (**e**) a composite knot consisting of the two trefoils joined with the 6-crossing knot $6_2$ (that is, $3_1^2 \# 6_2$), which passes through the periodic boundary of (**a**) many times with $L = 1{,}700\ \lambda$ and $|\Delta(-1)| = 99$; (**f**) the more complicated 14-crossing prime knot from (**b**) labelled $K14n5049$ in the extended notation of standard tabulations beyond 11 crossings[29], with $L = 1{,}500\ \lambda$ and $|\Delta(-1)| = 313$; (**g**) the open 8-crossing prime knot $8_{12}$ from (**c**) having $L = 1{,}000\ \lambda$ where $\lambda$ is defined with respect to momentum at the origin and only vortex length in the classically allowed region is considered, and with $|\Delta(-1)| = 29$.

Links, which are configurations of two or more vortex curves that are topologically entangled with one another, also occur frequently. In fact, we find links to be more common than knots, consistent with previous investigations of random optical fields at smaller lengthscales in which links are found to be common but knots were not detected[30]. The restriction to knotting in the present study was chosen as it allows comparison with extensive studies of the knotting probability of random curves (as random walks)[25], whereas the study of random linking is not so well developed. Furthermore, linking of open curves (in the 3DHO) is not well defined. For these reasons, our analysis is limited to the statistics of vortex knotting.

**Probabilities and complexities of knottedness.** As one might expect, the random eigenfunction statistics show that longer knotted vortex curves display a greater complexity of knot types, as measured by the knot determinant $|\Delta(-1)|$. This is represented in Fig. 2 for each of the three systems at fixed energy. These histograms indicate that the distribution of $\log |\Delta(-1)|$ with respect to vortex length $L$ apparently scales according to a power law. Longer curves are also more likely to be knotted; the probability of a given vortex loop being unknotted decreases exponentially according to $L$ (Fig. 2 insets), as previously studied for random walks modelling polymers[25,27,31]. The value of the unknotting exponents is different for each of the three systems, given in the caption.

Each of the model systems studied has a finite spatial extent, namely the side length of the cube and the diameters of the 3-sphere and classically allowed region of the 3DHO. In the latter two cases, this finite size imposes a long-length cutoff $S$ to the Brownian scaling of long vortex lines. In the 3-sphere and 3DHO, loops whose radius of gyration exceeds $S$ are confined, and the Brownian and confined regimes have different knotting probability exponents. Both knotting probability and complexity increase distinctly with length in the confined regime, where long curves are restricted to volumes much smaller than the corresponding Brownian radius of gyration would allow.

In contrast, the cube's periodic boundary conditions allow the vortex lines to have extent greater than the cube's side length, although the Brownian scaling of loop sizes gives way at some $S$ to the lines with nontrivial homology, which do not contribute to the knot count. Figure 2b,c shows that the minimal complexity of knots in the confined regime also scales with $L$; long vortices are always knotted in these systems, and the knot is usually very complex. By contrast, under periodic boundary conditions (Fig. 2a) even the longest vortex loops can be topologically trivial. In all of these cases, the scaling depends weakly on the energy of the eigenfunction, at least in the ranges of energy considered in the simulations.

**Low-energy knots and the impact of eigenfunction symmetry.** The total vortex length in random eigenfunctions is proportional

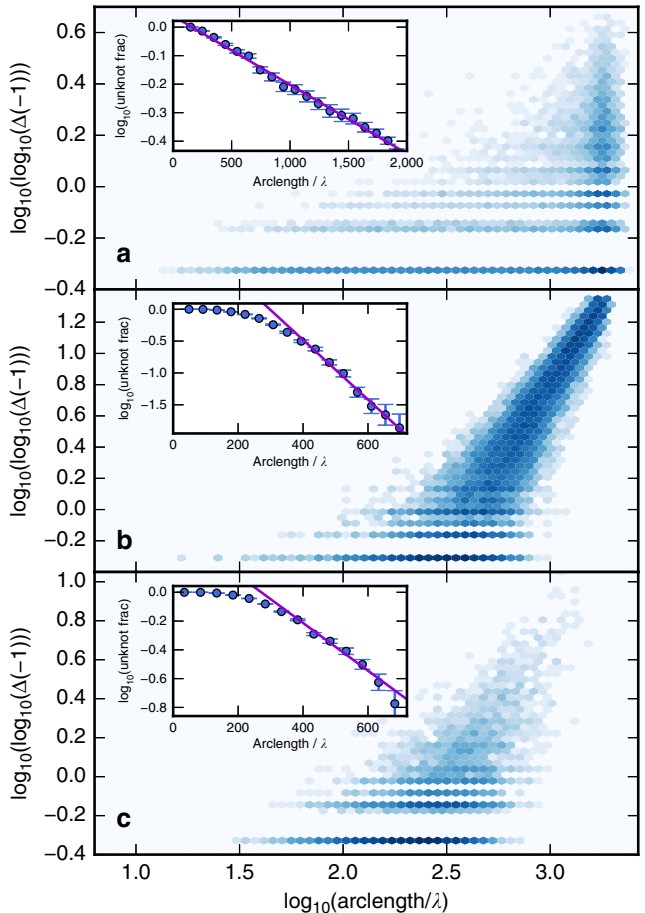

**Figure 2 | Dependence of vortex knot complexity on curve length.** The plots show histograms of $\log_{10}(\log_{10}|\Delta(-1)|)$ against $\log_{10}(L)$ for the thee systems at the same energies of the examples in Fig. 1: (**a**) periodic cube, $N=9$; (**b**) 3-sphere, $N=17$; (**c**) 3DHO, $N=21$. In each system, the logarithm of the probability that a given vortex curve is unknotted as a function of its length $L$ is plotted in the insets, with error bars representing the standard error of the mean. probability over many different random eigenfunctions. For larger values of $L$, this unknotting probability logarithm is fitted to $-L/L_0 + \mathrm{const}$, with scaling factors $L_0$ of approximately: (**a**) 1,800 $\lambda$; (**b**) 100 $\lambda$; (**c**) 250 $\lambda$. In the main plots, the strong horizontal lines of constant, low-value $|\Delta(-1)|$ correspond to specific knots, such as the trefoil (for which $\log_{10}(\log_{10}|\Delta(-1)|) = -0.32$) and the composite double trefoil $3_1\#3_1$ (for which $\log_{10}(\log_{10}|\Delta(-1)|) = -0.02$).

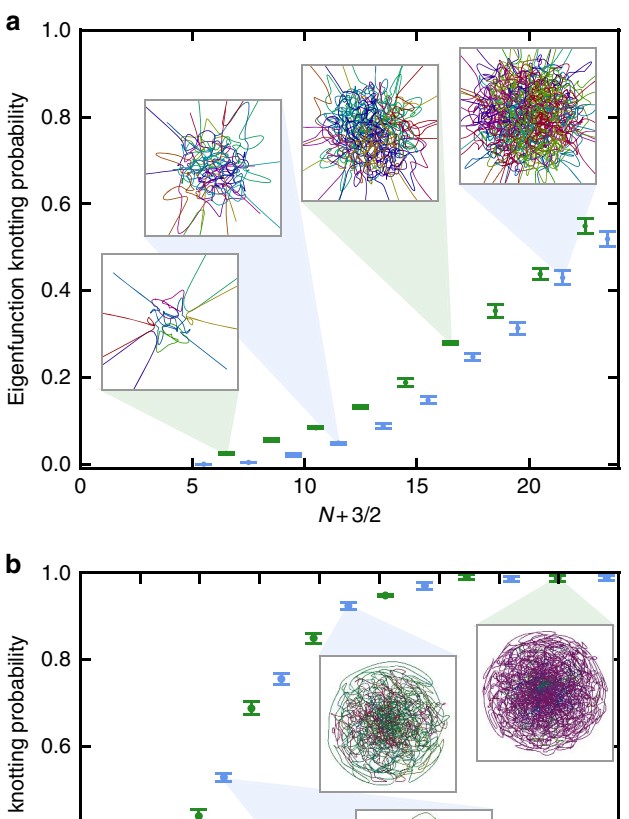

**Figure 3 | Probabilities of random eigenfunctions with different energies containing knotted vortices.** (**a**) 3DHO; (**b**) 3-sphere. Blue (green) points denote those where the principal quantum number $N$ is even (odd), for reasons described in the main text, and errors represent the standard error of the mean. fraction of knotted eigenfunctions, averaged over many different random samples. Insets depict the vortices in a typical eigenfunction at different energies, with each vortex curve represented in a different colour.

to their energy $E_N$ (see Supplementary Note 1). It is therefore natural to expect knotted vortex lines to occur more frequently in higher-energy eigenfunctions. This probability of knotting with energy is shown for eigenfunctions of the 3-sphere and 3DHO in Fig. 3. In the 3-sphere, where all vortex loops must be closed, this probability reaches 99% when $N \approx 16$, which is surprisingly small to guarantee such a degree of topological complexity (as illustrated in the insets of Fig. 3).

The energies simulated in the 3DHO are insufficient to guarantee that a random eigenfunction will contain a knotted vortex (Fig. 3a), as the distribution of vortex lengths in the classically allowed region prefers a greater number of shorter curves. The knotting probability of a 3DHO eigenfunction also strongly depends on whether the principal quantum number $N$ is odd or even, since 3DHO energy eigenstates, and hence the zeroes (that is, the vortex tangle), must be parity-symmetric with respect to spatial inversion through the origin; in fact, the nodal system of vortex curves is required to be strongly

negatively amphicheiral[32] (tangents at parity-opposite vortex points must be parallel). Furthermore, when $N$ is odd, exactly one open vortex line must pass through the origin, such as the low-energy example in Fig. 4a. Since strongly negatively amphicheiral knots must be open (see Supplementary Note 4 and Supplementary Fig. 6), this line can take such a configuration and is commonly knotted, contributing to a larger knotting probability when $N$ is odd (other knots can occur when $N$ is odd or even, but only in antipodal pairs, as shown in Fig. 4b). This symmetry constraint applies to all energies considered in the simulations.

Eigenfunctions of the 3-sphere must also be symmetric under inversion between antipodal points, but here the curve system must be strongly positively amphicheiral (opposite vortex points have antiparallel tangents). The simplest prime knot with this symmetry is in fact the 10-crossing knot $10_{99}$ (ref. 32), and an example of a knotted vortex of this type is shown in Fig. 4c.

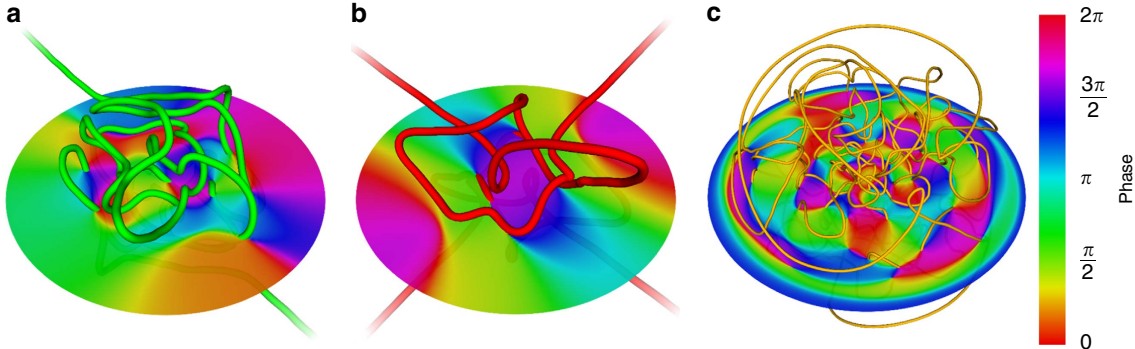

**Figure 4 | Examples of low-energy eigenfunctions with knotted vortices in symmetric strongly amphicheirally symmetric conformations.** (**a**) 3DHO, $N = 5$, with a single 'open figure-8' vortex knot (tabulated as $4_1$) in a strongly negatively amphicheiral conformation (this is the simplest prime knot with this symmetry); (**b**) 3DHO, $N = 4$, with a pair of mirrored trefoil vortex knots $3_1 \# 3_1$ (knotting at this low energy is found to be extremely rare); (**c**) 3-sphere, $N = 7$, with the knot $10_{99}$ (the simplest prime knot with strongly positively amphicheiral symmetry[32]), but omitting the two other small vortex loops occurring in this eigenfunction. In each case, the phase in a plane through the centre of the complex random wavefunction is represented, coloured by hue. Supplementary Movies 1–3 show these wavefunctions from varying viewpoints.

## Discussion

On the basis of computer experiments, we have shown that knotted vortex lines are common in random quantum wavefunctions, even at comparatively low energies. These results complement previous rigorous mathematical studies, where small knots (of any type) have been proven to occur with non-zero probability in random eigenfunctions of the 3DHO[33] and hydrogen atom[34] at high energies. These knots would be quite different from those studied in the present work, as they occupy limited volumes and, in the 3DHO, occur with exponentially small probabilities. As such, those knots may be thought of as highly structured superoscillatory phenomena[35], known to be very rare. On the other hand, the knotted vortices emphasized here can extend throughout the entire wavefunction, and are rather common except at the lowest energies; in fact, the very smallest knots we find in the 3DHO are an order of magnitude larger than the bounds on the size of superoscillatory examples. Preliminary numerical studies of vortex knots in random eigenfunctions of hydrogen indicate similar behaviour to the 3DHO described here, where eigenfunctions containing knotted vortices are typically beyond a certain energy (and comparable mode count), although unlike the 3DHO and 3-sphere, the nodal sets are not constrained by an amphicheiral symmetry.

Although our computer experiments have been limited to modes of fixed energy, we do not anticipate major differences in systems of random waves with different power spectra, such as a turbulent Bose–Einstein condensate[22] with characteristic power law spectrum, especially when sizes of system are similar to or greater than those considered here.

Knotting of vortices in chaotic complex 3D eigenfunctions is a complex counterpart to Chladni's original observation of the structure of cavity modes, and we anticipate that generic 3D complex cavity modes[1,2] will contain knotted vortices. Our results lead us to expect that almost every random quantum eigenfunction at sufficiently high energy contains at least one knotted vortex line and, at least in the systems considered here, this is common even at the relatively low energies currently accessible to numerical experiment. Knotted nodal lines with characteristic complexity scalings may therefore be a complex, 3D counterpart to the nodal statistics proposed to signify quantum chaos[36] for real-valued chaotic eigenfunctions in two dimensions. On the basis of the mode counts[2] of the energies in Fig. 3, we expect knotted vortices to occur with 50% probability from somewhere between the 500th (3-sphere) to 3,000th (3DHO) mode of a chaotic cavity.

## Methods

**Sampling random eigenfunctions.** We generate random eigenfunctions of model systems via standard random superpositions of their degenerate eigenfunctions. Further details and comparisons of these systems, especially their eigenfunctions, are included in Supplementary Note 2.

**Tracking vortices (nodal lines).** Within each random eigenfunction, phase vortices are numerically tracked by sampling the wavefunction on a cubic lattice (initial voxel size $\sim 0.1\lambda$) and checking around each numerical grid plaquette for the circulation of phase indicating the passage of a vortex line through the face of the grid cell. The phase is only guaranteed to circulate in this way when the amplitude is zero but the complex scalar gradient is non-zero (that is, the vortices occur as transverse intersections of real and imaginary nodal surfaces), but this is generically the case in random eigenfunctions; the algorithm would otherwise fail to converge on the vortex line but, as anticipated, we have never observed this to happen. This core procedure does not always fully capture the geometry of the vortex curve (especially where two vortex lines approach closely), and so the above procedure is augmented with a recursive resampling of the numerical grid, at successively higher resolutions, in any location where the vortex line tracking is ambiguous; we call this the RRCG algorithm (see Supplementary Note 2). Since vortex lines must be continuous, the result of the algorithm is a piecewise-linear representation of each vortex in the entire large-scale tangle. The vortex topology is retained, and geometry well recovered, even with a relatively poor initial sampling resolution. The algorithm generalizes to each of periodic boundary conditions, the 3-sphere and the 3DHO via an appropriate choice of lattice boundary conditions, and in the case of the 3-sphere by sampling on multiple Cartesian lattices and joining appropriate boundaries to reproduce the geometry of the sphere.

**Detecting and classifying knots and links.** The topology of vortices is investigated using the standard knot invariants described in the main text, whose values depend only on the topology of a single curve and from which the knot type can be inferred. It is computationally expensive to use some invariants known to be more powerful (for example, the Jones or HOMFLY-PT polynomials), or to identify the exact knot type of every curve. When necessary (Figs 1 and 4) we are able to do so unambiguously using the Alexander polynomial (evaluated at certain roots of unity), hyperbolic volume and Vassiliev invariants of degree 2 and 3. We further use the knot determinant to measure the topological complexity of vortex curves even without identifying their specific knot types.

**Data availability.** All relevant data are available from the corresponding author on request.

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

## Acknowledgements

We thank Michael Berry for valuable discussion and for originally proposing the problem investigated here, and Andrea Aiello, Stu Whittington, Daniel Peralta-Salas and Dmitry Jakobson for useful suggestions. This research was funded by a Leverhulme Trust Research Programme Grant. This work was carried out using the computational facilities of the Advanced Computing Research Centre, University of Bristol. We are grateful to the KITP for hospitality during the early part of this work. Alexander Taylor was partially funded by the Engineering and Physical Sciences Research Council and Mark Dennis by the the Royal Society of London.

## Author contributions

A.J.T. created the numerical routines for simulating eigenfunction modes and performing topological analysis. Other contributions were shared equally.

## Additional information

**Competing financial interests:** The authors declare no competing financial interests.

