## [Peer review file · Nature Communications]

Reviewers' Comments:

Reviewer #1 (Remarks to the Author)

In the paper under review, the authors study the nodal sets of high energy complex-valued eigenfunctions in three systems: the cube with periodic boundary conditions, the 3D isotropic harmonic oscillator and the 3-sphere. Generically, such nodal sets are smooth curves, and as such they can be knotted and linked. The authors analyze the energies and lengthscales at which knotting of these lines occurs and the probability of knottedness. This is a very interesting work, containing remarkable and deep results. To study the nodal lines, the authors perform a very fine numerical analysis of random eigenfunctions. In particular, this article gives rise to several conjectures that will be the object of desire of many physicists and mathematicians in the following years, e.g. does the probability of knottedness tends to 1 as the energy of the eigenfunction tends to infinity? In summary, I strongly recommend this paper for publication in Nature Communications. I have a few remarks that the authors should address before publication: (1) the authors talk about vortex filaments, they should explain what this means in the context of quantum mechanics, indeed in the context of fluid mechanics a vortex filament is a region of high concentration of vorticity, but it seems to be the opposite in quantum mechanics (the quantum probability density is zero on the nodal line); (2) They also talk about chaotic eigenfunctions, so the authors should explain the relevance of this work for the theory of quantum chaos, I did not find any clear explanation of this in the text; (3) The only rigorous theorem on the realization of knotted nodal lines of arbitrary topology in the 3DHO system is Ref. [29] cited in the paper, where the knots that are constructed appear at lengthscales of order $E_N^{-1/2}$ for high energies (so the knots are very small), the authors should comment if they also find knots/links at this scale in contrast with the aforementioned reference; (4) There is an extremely important issue that is not addressed by the authors, i.e. the structural stability of the knots they find, that is if the intersection of the real and imaginary parts of the quantum eigenfunction on the nodal line is transverse, thus guaranteeing that the phenomenon has non-zero measure and is physically observable; since the authors can detect the knotted structures in a computer, I guess that this should be the case, but it has to be checked; (5) Finally, the authors always talk about knots, but it seems to me from the computer simulations that links also appear, so a comment on this should be added.

Reviewer #2 (Remarks to the Author)

This is an excellent paper. The results about knottedness of vortex loops in numerous systems are very stimulating. I recommend the paper be published as it is.

We are grateful to the anonymous referees for their positive response to our manuscript, and particularly for the helpful comments of Referee #1. Below are our point-by-point responses to these queries, including details of other scientific changes made to the manuscript.

>"(1) the authors talk about vortex filaments, they should explain what this means in the context of quantum mechanics, indeed in the context of fluid mechanics a vortex filament is a region of high concentration of vorticity, but it seems to be the opposite in quantum mechanics (the quantum probability density is zero on the nodal line)"

Following the requirements of the journal, the introduction has been rearranged, and we now describe the connection between quantum mechanics and hydrodynamics via Madelung's interpretation of 1927. In particular, a major difference in quantum wavefunctions is that the vorticity is infinitely concentrated on the nodal lines, and zero everywhere else. This vorticity (as the net phase change around a vortex) is quantized in multiples of 2π , and we briefly discuss these points in the new introduction. In answer to the referee's point, although the quantum probability density vanishes at the vortex core, the phase change is infinite and requires regularization (discussed in the literature on quantum vortices).

>"(2) They also talk about chaotic eigenfunctions, so the authors should explain the relevance of this work for the theory of quantum chaos, I did not find any clear explanation of this in the text"

We now describe wave chaos further in the introduction, motivating our work as a study of wave chaos via quantum eigenfunctions. The high-energy limit of all our model systems is the isotropic random wave model, the model eigenfunction for quantum chaotic systems, but our study is not directly about quantum chaos, as we are not studying any particular wave ergodic systems (such a system could be a 3D Bunimovich stadium, for instance). Nevertheless, given the connection between models of wave chaos and quantum chaos, we elaborate our conjecture on knotted vortices in chaotic cavity modes in the final section.

>"(3) The only rigorous theorem on the realization of knotted nodal lines of arbitrary topology in the 3DHO system is Ref. [29] cited in the paper, where the knots that are constructed appear at lengthscales of order $E_N^{-1/2}$ for high energies (so the knots are very small), the authors should comment if they also find knots/links at this scale in contrast with the aforementioned reference"

The knots found in our computer experiments are all much larger than the small knots shown to exist at sufficiently high energies by Enciso, Hartley and Peralta-Salas in the 3DHO. We now directly compare the two kinds of systems, identifying the rigorous results as what are known as ‘structured superoscillations’ in wave physics, which are expected to be exponentially much rarer than the knots we describe. We have also added a new reference to more recent work by the same authors on vortex knots in hydrogen wavefunctions, and a brief comparison to our own preliminary analysis of this system (with details omitted).

>"(4) There is an extremely important issue that is not addressed by the authors, i.e. the structural stability of the knots they find, that is if the intersection of the real and imaginary parts of the quantum eigenfunction on the nodal line is transverse, thus guaranteeing that the phenomenon has non-zero measure and is physically observable; since the authors can detect the knotted structures in a computer, I guess that this should be the case, but it has to be checked"

The structural stability of nodal lines in complex scalar wavefunctions is well established. The fact these structures arise statistically in our studies shows they should occur with nonzero measure; indeed, this is the main result of our computer experiments. We have added some comments in the methods describing how the numerical vortex tracking would fail in the presence of nongeneric structures (codimension 4 examples being nodal points and crossings of vortex lines). As one might expect, we have never observed such events in the numerical eigenfunctions, on looking with various methods.

>"(5) Finally, the authors always talk about knots, but it seems to me from the computer simulations that links also appear, so a comment on this should be added"

We are grateful to the referee for this observation! Links do occur, and are even more common than knots in the systems we study. However, linking is a less convenient tool to compare the topology of vortex curves: the statistical topology of linked random walks is far less well established than knotting (which provides us with a major basis of comparison), and there is no convenient complexity measure for links (linking number itself being insufficient to distinguish many complex links). Furthermore, linking of two open curves cannot be defined, limiting the study in the 3DHO. We have added a paragraph mentioning how links are common, but briefly explaining our reasons for limiting the present study to knots alone.

Reviewer #1 (Remarks to the Author)

[The referee recommends the paper for publication with no further comments for the authors]